# The Effects of COVID-19 on Placenta and Pregnancy: What Do We Know So Far?

**DOI:** 10.3390/diagnostics11010094

**Published:** 2021-01-08

**Authors:** Yin Ping Wong, Teck Yee Khong, Geok Chin Tan

**Affiliations:** 1Department of Pathology, Faculty of Medicine, Universiti Kebangsaan Malaysia, Jalan Yaacob Latif, Bandar Tun Razak, Kuala Lumpur 56000, Malaysia; 2Department of Pathology, SA Pathology, Women’s and Children’s Hospital, North Adelaide, SA 5006, Australia; yee.khong@adelaide.edu.au

**Keywords:** COVID-19, pathology, placenta, pregnancy, SARS-CoV-2

## Abstract

The current coronavirus disease 2019 (COVID-19) pandemic, caused by the novel severe acute respiratory syndrome coronavirus 2 (SARS-CoV-2), has inflicted a serious health crisis globally. This virus is associated with a spectrum of respiratory illness ranging from asymptomatic, mild to severe pneumonia, and acute respiratory distress syndrome. Accumulating evidence supports that COVID-19 is not merely a respiratory illness per se, but potentially affects other organ systems including the placenta. SARS-CoV-2 gains access to human cells via angiotensin-converting enzyme 2 (ACE-2). The abundance of ACE-2 on the placental cell surface, especially the syncytiotrophoblasts, could potentially contribute to vertical transplacental transmission to the fetus following maternal COVID-19 infection. Intriguingly, despite the placentas being tested positive for SARS-CoV-2, there are very few newborns that manifest virus-induced diseases. The protective effects of the placental barrier to viral infection, limiting the spread of the virus to newborn infants, remain a mystery. The detrimental role of COVID-19 in pregnancies is largely debatable, although COVID-19 maternal infection has been implicated in unfavorable pregnancy outcomes. In this review, we summarize the pathological features manifested in placenta due to COVID-19 maternal infection that have been previously reported, and relate them to the possible disease manifestation. The potential mechanistic pathways associated with transplacental viral transmission and adverse pregnancy outcomes are also discussed.

## 1. Introduction

The 2019 coronavirus disease (COVID-19), a novel zoonotic disease [1], was first discovered in late December 2019 following an outbreak of severe pneumonia of unknown etiology in Wuhan, Hubei Province, China. The etiological agent was successfully isolated and identified as a previously unknown beta-coronavirus, which was provisionally coined as 2019 novel coronavirus (2019-nCoV) [2]. It was later officially designated as Severe Acute Respiratory Syndrome Virus 2 (SARS-CoV-2) on the ground of phylogenetic analysis by the International Committee on Taxonomy of Viruses [3]. The emergence and rapid spread of SARS-CoV-2 via sustained human-to-human transmission poses a formidable pandemic threat to humankind globally. COVID-19 was declared as the fifth documented pandemic on 11 March 2020 by the World Health Organization (WHO) after the 2009 Influenza A (H1N1) swine flu [4]. As of 9 November 2020, there were estimated to be 50.4 million confirmed cases worldwide with 1.26 million deaths, accounting for an almost 2.5% death rate [5].

SARS-CoV-2 infection has taken healthcare providers by surprise, as it behaves like no other “respiratory” infection, and lungs are not the only organs being affected. A proportion of patients with severe COVID-19 presented with extrapulmonary clinical manifestations related to cardiac-, kidney-, liver-, digestive tract-injuries, and neurological disorders [6], besides suffering from classic respiratory symptoms and fever. Placentas are no exception. There is increasing evidence that COVID-19 infection leaves tell-tale signs of injuries in the placenta.

Interestingly, despite the increasing molecular and ultrastructural evidence of SARS-CoV-2 in the placentas of COVID-19-positive mothers, newborns have yet to manifest virus-induced diseases [7]. No teratogenic effect of COVID-19 infection in the neonate has been reported. Gajbhiye et al. (2020) observed that only 24 (8%) out of 313 neonates born to mothers with COVID-19 tested positive for SARS-CoV-2 [8], which raises an important question on the success rate of transplacental viral infection (intrauterine transmission) to the fetus. Noteworthy, maternal infection does not equate to placental infection. Likewise, evidence of placental viral infection does not guarantee intrauterine vertical transmission to fetus [9]. It is assumed that there will be an active replication of the virus in the placenta. However, if this is true, the mechanism involved in preventing this highly infectious virus from reaching the fetus is still unclear. Possibilities include the maternal–fetal interface of placenta acting as a strong barrier against infection, or the absence of specific pathways/receptors that allow effective viral transmission.

The human placenta has an immunological barrier to the entry of pathogens, besides maintaining immune tolerance to the fetal cells. The key role of innate immune system in protecting the fetuses and neonates against SARS-CoV-2 infection has been proposed [10]. Decidua basalis, being the maternal component of the maternofetal interface, contains diverse immune cells that belong to the innate immune system including natural killer (NK) cells (70%), decidual macrophages (15%), and CD4 T cells (15%) [11]. In addition, the syncytiotrophoblast cells, outermost layer of chorionic villi, which are in direct contact with the maternal blood, do not contain an intercellular gap junction, and thus prevent entry of pathogens from the maternal blood. Physical obstacles including trophoblastic basement membranes create an additional physical barrier against pathogens [9]. Taken together, the innate immune system, structural barrier, as well as the interaction between decidual immune cells and the invading fetal extravillous trophoblasts may play a role in the placental protective mechanisms against SARS-CoV-2 viral invasion.

Unequivocal diagnosis of SARS-CoV-2 transplacental infection requires the detection of viral RNAs in placenta, amniotic fluid prior to the onset of labor, cord or neonatal blood/body fluid/respiratory samples, or demonstration of viral particles by electron microscopy, immunohistochemistry, or in situ hybridization method in fetal/placental tissues [12]. The usefulness of serological testing in the diagnosis of SARS-CoV-2 infection is yet to be confirmed [13]. The specificity of IgM testing is questionable, due to high false positivity [14]. Alternatively, testing for viral load in both the maternal and fetal plasma (viraemia) in lieu of nasopharyngeal swab may be useful to determine if the risk of transmission positively correlated with maternal viral load, besides helping to estimate the transmission rate more accurately.

Placental examination can yield invaluable information that may be essential to enhance our understanding of disease pathogenesis and to identify underlying causes of adverse pregnancy outcomes [15]. Due to the novelty of COVID-19, histomorphological and ultrastructural changes reported on placentas from SARS-CoV-2-positive women are limited to isolated case reports and a handful of case series. The knowledge gap between the effects of COVID-19 on placenta and pregnancy outcomes needs to be explored.

With the aim to address this issue, we performed a literature search on the MEDLINE/PubMed electronic database using keywords “COVID-19”, “SARS-CoV-2”, and “placenta”. Reference lists of the included papers were also checked to identify additional relevant studies. We reviewed all articles (both peer-reviewed and preprints) with reports on placental pathology in pregnant women tested positive for SARS-CoV-2 that were published between 1 January 2020 to 10 October 2020 (*n* = 29) [16,17,18,19,20,21,22,23,24,25,26,27,28,29,30,31,32,33,34,35,36,37,38,39,40,41,42,43,44]. Articles written in languages other than English and Mandarin were excluded. Clinical characteristics of the SARS-CoV-2 included studies were summarized in Appendix A.

## 2. SARS-CoV-2 and Pregnancy

Coronaviruses (CoV) including Severe Acute Respiratory Syndrome (SARS)-Co-V, Middle East Respiratory Syndrome (MERS)-CoV, hCoV-HKU1, and hCoV-OC43 are among the known zoonotic viruses that cause respiratory and gastrointestinal infections in humans [45], with SARS-CoV-2 being the latest discovered. SARS-CoV-2, like its predecessors SARS-CoV and MERS-CoV, is highly pathogenic and lethal, causing severe pneumonia, acute respiratory distress syndrome (ARDS), multi-organ failure, and death [46].

SARS-CoV-2 is a large, spherical, enveloped, single stranded positive-sense ribonucleic acid (RNA) virus with genome size approximately 30 kilobase (kb) in length. There are four major structural proteins that form the backbone of the virus: The spike (S) protein, membrane (M) protein, envelope (E) protein, and nucleocapsid (N) protein. The unique structural protein, the spike (S) protein, which is present in abundance on the viral cell surface, plays a key role in its pathogenesis [47].

Pregnancy has always been associated with an increased risk of acquiring respiratory infection with higher morbidity and mortality than the nonpregnant subjects [48]. In a systematic review examining 2567 COVID-19 confirmed pregnancies with 746 deliveries, there were 3.4% women with maternal critical disease requiring mechanical ventilation support, 0.9% maternal death, and 21.8% preterm birth, primarily iatrogenic rather than spontaneous, and less than 1% perinatal death [49]. In another systematic review on 324 pregnancies with COVID-19, it was reported that up to 14% mothers with severe pneumonia required critical care, with a total of 9 cases of maternal deaths, 4 cases of spontaneous abortion, 4 cases of intrauterine fetal deaths, and 3 cases of neonatal death [50]. Brandt et al. (2020) conducted a matched case-control study involving 61 COVID-19 pregnancies matched to 2 controls by delivery date, and reported that the odds of adverse composite outcomes for mother and fetus were 3.4 times and 1.7 times higher among severe/critical COVID-19 cases than controls, respectively, with maternal advance age, obesity, Hispanic or Latino origin, and other medical comorbidities being some of the important factors driving a more severe clinical course. Reassuringly, pregnant women who were asymptomatic or had mild COVID-19 disease were observed to have benign outcomes [51].

The anatomical and physiological changes of the respiratory system as well as immunological and hormonal adaptation during pregnancy collectively make them more vulnerable to certain infections, including SARS-CoV-2 [52,53]. Anatomical and physiological changes in the maternal respiratory system include relaxation of rib ligaments, diaphragmatic elevation, and reduction in the functional residual capacity (FRC) of the lung initiated by the effects of progesterone, and hence ineffective airway clearance [53]. Functional ventilation and perfusion mismatch as a sequela of reduced FRC also increases the severity of respiratory infection. Besides, immunological modulation by a physiological shift to Th2 dominant environment and attenuation of Th1 cell-mediated immunity during pregnancy has increased viral infection risks. It does not only affect the viral clearance rate, but hastens the disease deterioration [54].

Notably, SARS-CoV-2 infects target host cells by binding to the cell membrane angiotensin-converting enzyme II (ACE2), facilitated by S protein priming proteases Type II transmembrane serine protease (TMPRSS2). ACE2 is a membrane-bound aminopeptidase enzyme that has a physiological role in degrading substances including angiotensins I and II, a key protective mechanism in regulating vascular and heart functions. It is found in most organs such as heart, lung, kidney, vessels, brain, and others, including the placenta [55]. The types of cell in placenta that express ACE2 are the syncytiotrophoblasts and cytotrophoblasts in villi, decidual stromal cells, decidual perivascular cells, and endothelial and vascular smooth muscle cells. The co-expression of the viral receptor ACE2 and TMPRSS2 in abundance in the human placenta theoretically may increase vulnerability of placenta and possibly fetus to SARS-CoV-2 infection [56,57].

Interestingly, subsequent studies pursued by Bloise et al. (2020) revealed that ACE2 and TMPRSS2 were differentially expressed at different stages of pregnancy. They observed that ACE2 and TMPRSS2 expressions in placenta were negatively correlated with gestational age, in which their expression levels in reducing trend from first trimester to second trimester placenta, and very low to almost undetectable in third trimester preterm and term placenta [57]. The authors concluded that the first trimester of pregnancy was possibly more vulnerable to SARS-CoV-2 transplacental transmission than in later stages of pregnancy [57]. Surprisingly, SARS-CoV-2 was not detected in the abortus/placenta in the few reported cases of first and second trimester miscarriages despite the mothers being tested positive for COVID-19 [19,58], but was found to be positive in amniotic fluid and infant born pre-term to a COVID-19-positive mother in the third trimester [59], even though, theoretically, third trimester infants should be protected from acquiring COVID-19 infection since there is a paucity of the required receptors present in the placenta.

In addition, Kotlyar et al. (2019), in their systematic review revealed that SARS-CoV-2 positivity detection rate from neonatal nasopharyngeal swab, cord blood, placenta samples, amniotic fluid, and serology was extant although low (ranges from 0–7.7%) [60]. It is possible that there is another alternate entry of SARS-CoV-2 into placenta cells beyond ACE2. Alternatively, it is possible that the presence of underdiagnosed/asymptomatic concurrent genital tract infection breaches the placental barrier and allows entry of the virus into amniotic cavity.

Viraemia is a prerequisite for transplacental transmission to occur. Emerging evidence showed that plasma viral load in COVID-19 cases was positively correlated with the disease severity [61]. Infants born to mothers with severe to critical COVID-19 diseases showed higher rates of SARS-CoV-2 positivity detected in the placenta and nasopharyngeal swab immediately after birth [26,62], supporting the viraemia hypothesis. Overall prevalence of viraemia in COVID-19 infected individuals, however, is low and transient [63], which makes placental seeding rather impossible and hence, the low rate of transplacental transmission. Arguably, the mere detection of viral RNA in the blood/serum does not reflect infectivity of a virus. Andersson et al. (2020) found that all their reverse transcription-polymerase chain reaction (RT-PCR)-positive clinical samples show neither viral replication nor display any cytopathic effect after inoculation in cell culture [64]. More studies are warranted to resolve these uncertainties.

Other possible routes of intrauterine transmission include ascending route via vaginal secretion by which the microorganisms gain access to uterine cavity and infect the amniotic fluid. However, studies revealed that the samples isolated from vaginal environment, which includes vaginal secretion and cervical exfoliated cells, all returned negative results [65].

Since plasma viral load in COVID-19 cases was positively correlated with the disease severity, it can serve as an excellent biomarker to predict impending critical illness [61]. Viraemia, albeit transient, may trigger a “cytokine storm” following overwhelmed maternal immune response to SARS-CoV-2 infection with a surge in pro-inflammatory cytokines including interferon-ɣ (IFN-ɣ), interleukin-2 (IL-2), IL-6, IL-7, IL-10, and tumor necrosis factor-α (TNF-α). Others believed that the increased production of inflammatory biomarkers may be triggered by placental hypoperfusion/ischemia due to maternal hypoxia following severe COVID-19 infection [66]. The uncontrolled release of inflammatory cytokines will further exaggerate the maternal immune system and link to the occurrence of placental damage, fetal growth restriction, abortion, or preterm labor [67,68]. Elevated IL-6 and TNF-α levels induce endothelial dysfunction, a hallmark of pre-eclampsia and predispose to a maternal thromboembolic event [41]. Some suggested that inactivation/downregulation of ACE2 occurs via formation of viral-ACE2 complex following SARS-CoV-2 placental infection causing lowering of plasma angiotensin-(1-7) levels, which in return potentiates vasoconstriction and pro-coagulopathic state, leading to early onset, severe pre-eclampsia [69].

Fetal inflammatory response (FIRS), as demonstrated by increased IL-6 following maternal COVID-19 infection [70], may be responsible for a wide range of adverse neurodevelopmental sequelae including autism, psychosis, and neurosensorial deficits later in life [71], similar to certain bacterial infections [72]. However, long term longitudinal studies are required to validate these associations.

Apart from transmissibility of the virus, in silico studies suggest that placental function may be modulated by SARS-CoV-2 via proteins other than ACE2 and TMPRSS2, such as those that are involved in trophoblast invasion and migration, syncytialization, and implantation [73]. How these other proteins correlate with the placental morphology so far described in COVID-19 cases and pregnancy outcomes remain to be determined.

## 3. Placental Morphological Changes

To date, there are 29 studies reporting the histopathological findings on six second trimester and 322 third trimester placentas from pregnant women diagnosed with COVID-19 infection of different levels of severity. There are no pathognomonic histological patterns in human placentas following SARS-CoV-2 maternal infection, as illustrated in a recent review by Sharps et al. (2020) on 20 studies [74]. The placentas that were analyzed were obtained from COVID-19 infected mothers regardless of the status of placental infection by SARS-CoV-2. The morphological changes described may not be proven significant considering a plethora of confounders. To validate the histomorphological findings in COVID-19 infected placenta with confidence, we reviewed only the reports which demonstrated at least the evidence of SARS-CoV-2 in the placenta cells/tissues by currently accepted standards. There was a total of 17 studies included (Appendix A), with 3 and 33 SARS-CoV-2-positive second and third trimester placentas, respectively. A summary of histopathological characteristics of COVID-19-infected placentas is depicted in Figure 1.

### 3.1. Histomorpholological Alterations

#### 3.1.1. Term and Preterm Third Trimesters

A higher frequency of maternal vascular malperfusion (MVM) of the placental bed was reported in placentas of pregnant women infected with SARS-CoV-2 by 13 studies [16,22,23,25,27,28,31,32,38,40,42,43,44]. It is a recognized pattern of placental injury related to abnormal uterine perfusion, leading to a myriad of pathological changes such as accelerated villous maturation, increased perivillous and intervillous fibrin deposition, decidual vasculopathy, Tenney–Parker change, villous infarction, and intervillous thrombosis [75]. It is associated with significant clinical sequelae, e.g., preterm birth, fetal growth restriction, and fetal demise. Maternal hypoxia secondary to severe COVID-19 lung infection may initiate uterine underperfusion and subsequent hypoxic-ischemic injury to the placenta. Taglauer et al. (2020) reported that 14 of the 15 (93%) third trimester placentas revealed at least a feature of MVM, with infarcts and increased fibrin deposition being the most frequently observed, compared to healthy controls where MVM was observed only in 30% (3/10) of cases [42]. This was similar with that reported by Zhang et al. (2020) and Hecht et al. (2020) [25,44], although their findings did not meet statistical significance.

Fetal vascular malperfusion (FVM) is a feature manifested in placentas with diminished vascular supply, reported in four studies with COVID-19-affected placentas [22,23,31,40]. Fetal vascular thrombosis, abnormal cord insertion, hypercoiling of umbilical cord, and maternal hypercoagulable state are among the conditions associated with FVM [76,77]. The endotheliotropic behavior of SARS-CoV-2 via the ACE2 receptor on endothelial cells makes it prone to cause vascular endothelial dysfunction, leading to a complement-induced coagulopathy state in COVID-19 infected patients and henceforth susceptible to microthrombi formation [78]. In addition, FIRS as a sequelae following severe COVID-19 maternal infection may be associated with endothelial or vessel wall damage, hence the possible pathogenesis behind the evidence of vascular thrombosis in fetal circulation [79]. Villous stromal-vascular karyorrhexis can be seen secondary to damage to/breakdown of fetal endothelial cells within villous stroma. Depending on the severity and the nature of obstruction, it poses a risk of fetal growth restriction, oligohydramnios, non-reassuring fetal heart rate, and fetal demise, as documented in one of the COVID-19-infected pregnancies [37].

Facchetti et al. (2020) observed that evidence of FVM such as avascular villi and villous stromal-vascular karyorrhexis was demonstrated in the SARS-CoV-2-positive placenta from a pregnant mother with known history of idiopathic thrombocytopaenia, who already had a pre-existing risk of thrombosis [22]. It is noteworthy to point out that the signs of MVM of the placental bed and FVM are non-specific and can be present in many maternal medical conditions such as hypertensive disorders in pregnancy [80], lupus anticoagulant, and protein C and protein S deficiency. The results should be interpreted with caution in the context of overall clinical scenarios, and may not be attributed to SARS-CoV-2 infection per se.

Similar to other RNA viral infections, e.g., Zika virus [81], cytomegalovirus [82], and dengue virus [83] in pregnancy, SARS-CoV-2 placental infection was associated with chronic inflammatory pathologies which include lymphohistiocytic villitis [27,31,40,42], chronic histiocytic intervillositis [22,28,32,35,38], and chronic deciduitis [31]. Unexpectedly, maternal inflammatory responses as characterized histologically by acute subchorionitis/chorionitis (stage 1) and chorioamnionitis (stage 2) were the most reported features in SARS-CoV-2 positive placentas (11/33, 33%), documented in 11 studies. In comparison, only one (1.7%) out of 60 SARS-CoV-2-negative placentas in COVID-19-infected mothers showed evidence of acute chorioamnionitis [30]. Fetal inflammatory response (chorionic vasculitis or fetal vasculitis) was occasionally described in SARS-CoV-2 positive placentas [25,31,42], but showed no statistical difference with the control group. Collectively, whether or not the presence and the extent of inflammatory response correspond with the detection of SARS-CoV-2 in placentas awaits further investigations.

An increase in intervillous macrophage infiltration is a consistent finding described in a few cases [22,28,32,35,38,40,43], in agreement with the autopsy finding of lung tissue from patients who died of severe COVID-19. The influx of macrophages in COVID-19 infected placenta is believed to help contain local viral propagation and transmission, but at the same time as a key mediator to exuberant immune response leading to unnecessary tissue destruction. While some believed that monocytes/macrophages could serve as vectors for viral dissemination [84,85], the presence of viral particles within the cytoplasm of fetal vascular monocytes may play a role in the spread of infection to the fetus. Robust therapeutic strategies to inhibit migration and differentiation of monocytes/macrophages might help thwart immunopathology associated to SARS-CoV-2 infection.

#### 3.1.2. Second Trimesters

So far, there are only five published studies describing the histopathological features of second trimester placentas (range from 16 weeks’ to 24 weeks’ gestation) obtained from COVID-19-infected pregnant mothers [19,26,37,38,39]. Placentas from two of the studies were not further tested for SARS-CoV-2 [38,39]. Hosier et al. (2020) revealed relatively high viral copy numbers in both the placenta (3 × 10^7^ virus copies/mg) and umbilical cord (2 × 10^3^ virus copies/mg), while the fetal heart and lung tissues tested negative [26]. Baud et al. (2020) found all fetal tissues, other than placenta tissue, and umbilical cord yielded negative COVID-19 results in a 19-week-old stillbirth [19].

Histologically, increased perivillous and subchorionic fibrin depositions were present in all three studies [19,26,37]. A 35-year-old COVID-19-infected pregnant woman presented with clinical abruption at 22 weeks’ gestation [26]. The author postulated that COVID-19-related placental inflammation may potentiate early-onset severe pre-eclampsia in a patient with previous history of gestational hypertension. Further studies are required to address this association [60]. Chronic histiocytic intervillositis with associated ischaemic necrosis of surrounding villi was displayed in placentas of a 24-week-old dichorionic diamniotic twin born to a 30-year-old mother with gestational diabetes mellitus [37]. Other findings include acute subchorionitis and fetal vasculitis, which were demonstrated in a 19-week-old stillbirth born to a 28-year-old COVID-19-positive mother who presented with preterm labor [19].

### 3.2. Immunolocalisation of Viral Proteins in Placentas

Various methods were employed to demonstrate the existence of virus. Immunolocalization of SARS-CoV-2 viral spike protein mRNAs by in situ hybridization (ISH) or viral nucleocapsid protein by immunohistochemistry (IHC) are among the preferred methods besides the standard RT-PCR method to detect viral RNAs.

Of the 29 studies with placental histopathological correlations, 14 studies had attempted to localize SARS-CoV-2 within placentas via IHC (*n* = 6) [16,27,32,37,40,43], ISH (*n* = 4) [31,35,42,44], or both (*n* = 4) [22,25,33,41]. Viral nucleocapsid protein and spike protein mRNAs were successfully identified in 17.4% (30/172) placentas, most commonly expressed in syncytiotrophoblasts [22,25,32,35,37,40,42,43,44] with rare expression in cytotrophoblasts [25,42], stromal cells [32], chorionic villi endothelial cells [27], and intervillous mononuclear cells [22]. In addition, rare viral mRNAs positivity detected by ISH in decidual endothelial cells [25,31] and endometrial glands [44] were also reported. Of the 30 positive placenta samples with babies that were subsequently tested for SARS-CoV-2, 40.0% (*n* = 12) were tested positive within 24 h after birth.

Detection of SARS-CoV-2 RNA via RT-PCR method was reported in 11 studies [16,19,20,23,26,28,30,32,37,38,43]. Among the 11 studies, additional tissues such as umbilical cords [20,26,28], cord blood [19,31], and amniotic fluid samples [19,31,37,38,43] were also tested. Successful detection of SARS-CoV-2 viral RNA by RT-PCR was reported in 9 studies [16,19,23,26,28,32,37,38,43], which included a total of 10 placentas (*n* = 14, 71.4%), 1 umbilical cords (*n* = 5, 20%), and 3 amniotic fluid samples (*n* = 5, 60%). Among these positive samples, only 20% (2/10) of the respective neonates were subsequently tested SARS-CoV-2 positive in the nasopharyngeal swab less than 24 h after birth [28,43].

### 3.3. Ultrastructural Analysis of SARS-CoV-2-Infected Placentas

There are four studies that explored the presence of SARS-CoV-2 viral particles with transmission electron microscopy. Hosier et al. (2020) was first to report the detection of viral particles within the cytosol of “placenta cells” in the placenta of a 35-year-old woman whose pregnancy was complicated with severe pre-eclampsia at 22 weeks of gestation [26]. The “placenta cells” described were later identified as syncytiotrophoblasts, microvilli, and cell process of fibroblasts in terminal villi in a placenta from a 28 year-old woman who presented with severe COVID-19 pneumonia at 28 weeks of gestation [16]. Subsequently, 89 to 129 nm-sized particles within membrane bound cisternal spaces sized, identified as SARS-CoV-2 viral particles, were found localized to syncytiotrophoblast cells in a 34-year-old woman who presented with preterm premature ruptured of membrane [40]. Other than syncytiotrophoblasts, fibroblasts, microvilli, and virions were found localized in fetal endothelial capillary cells close to villous surfaces as well as intravascular mononuclear cells, maternal macrophages, and Hofbauer cells [22]. To our knowledge, there are no published data of ultrastructural examination on placentas evaluating the ultrastructural alterations following SARS-CoV-2 infection.

## 4. Conclusions and Future Prospects

There is still plenty to learn about the virus and the detrimental effects it may bring to infected mothers and fetuses. Accumulating evidence suggests that intrauterine vertical transmission to fetuses does occur, albeit rarely. Although most neonates born to SARS-CoV-2 infected mothers do not seemed to acquire the infection postnatally, cases with neonates complicated with viraemia and subsequent neurological compromise have been reported [43,86]. Hence, the potential of SARS-CoV-2 affecting the newborns exists. Whether or not it is an immune-mediated event or direct cytopathic effect of the virus awaits further studies. There are no reported teratogenic effects of the virus on fetuses thus far. Detailed neonatal outcomes following maternal COVID-19, however, are beyond the scope of this review. Figure 2 summarizes the maternal–fetal interplay following SARS-CoV-2 infection, the reported histomorphological alterations in infected placenta, and its potentially associated adverse pregnancy outcomes.

The histological features of SARS-CoV-2-infected placenta are not well defined hitherto, although a few consistent histopathological abnormalities were observed. Arguably, chronic changes related to MVM and FVM, such as infarction, distal villous hypoplasia, and chorangiosis, require time to evolve and may not be manifested in acute COVID-19-infected placentas. Large scale pooled data to compare and contrast histological manifestations of COVID-19-infected and non-COVID-19-infected placentas at different phases of infection may be necessary to conclude the significance of the SARS-CoV-2-related histomorphological changes in placentas.

Nonetheless, there are a few limitations in our review. Although, a total of 29 studies with 6 s trimester and 322 third trimester placentas were included, these were limited to case reports, small studies, and very few large case studies. The placental histopathological information was mainly confined to placentas from third trimester. To the best of our knowledge, the detection of SARS-CoV-2 has not yet been described in products of conception of first trimester pregnancy. The protocol used to examine and sample placenta was not mentioned in most studies, although the histopathological findings reported were defined using the Amsterdam Consensus Criteria in the current review [15]. For instance, increased in fibrin deposition, a feature that was frequently documented in COVID-19-infected placentas, was not quantified in most studies. Redline et al. (1994) highlighted the significance of the extent and site of intervillous fibrin deposition, concluding that fibrin deposition around more than 20% of terminal villi in the mid and basal portions of placenta may interfere with feto-maternal perfusion, resulting in fetal growth restriction. Deposition of fibrin around the peripheral region of placenta, where less gas exchange takes place, has no clinical significance [87]. In addition, the cut off for the amount of perivillous, subchorionic, and intervillous fibrin deposition is arbitrary and not well defined. The exact location and extent of the intervillous fibrin deposition were not well described in the respective case series, hence direct association cannot be established with confidence.

In conclusion, the cumulative published data on placenta with COVID-19 infection showed common histological features, which include MVM, inflammation, and FVM. The risks of adverse perinatal and long-term outcomes of COVID-19 infection include miscarriage, stillbirth, fetal growth restriction, early onset pre-eclampsia, FIRS, and neurosensorial developmental delay. A well-designed prospective cohort study on larger population samples with thorough placental examination adopting standardized diagnostic approach is required to facilitate the current understanding of the impact of SARS-CoV-2 on the placenta and how it influences pregnancy outcomes. A causal relationship between maternal SARS-CoV-2 infection and placental pathology/pregnancy outcomes is essential to direct (and ensure) optimal maternity care in the challenging period of a global pandemic.

## Figures and Tables

**Figure 1 diagnostics-11-00094-f001:**
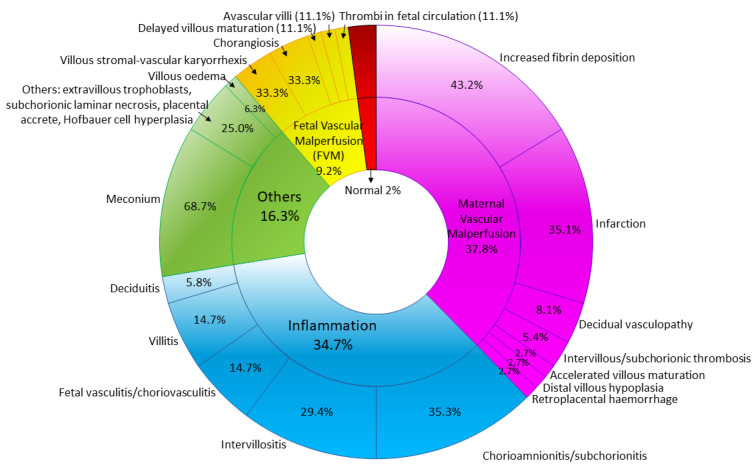
Frequency of reported histopathological features of. SARS-CoV-2-infected second and third trimester placentas (*n* = 36).

**Figure 2 diagnostics-11-00094-f002:**
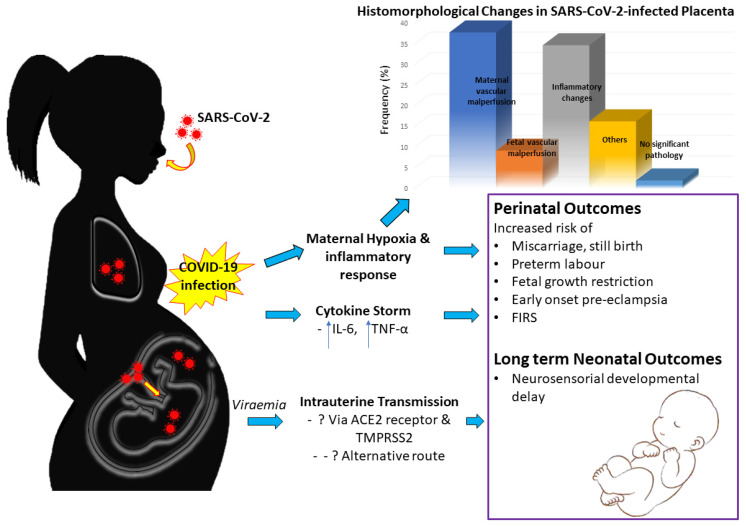
Maternal–fetal interplay following SARS-CoV-2 infection, reported histomorphological alterations in infected placenta, and adverse pregnancy outcomes. Abbreviations: ACE2: Angiotensin converting enzyme 2; FIRS: Fetal inflammatory response syndrome; IL-6: Interleukin-6; TNF-α: Tumor necrosis factor-α; TMPRSS2: Transmembrane serine protease 2.

## Data Availability

Not applicable.

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
