# Peer review of "The Effects of COVID-19 on Placenta and Pregnancy: What Do We Know So Far?"

_diagnostics, 2021, doi:10.3390/diagnostics11010094_

Round 1

Reviewer 1 Report

The authors performed a very extensive review of the existing literature on the placental effects of SARS-CoV 2 infection acquired in pregnancy. The proposed documentation is very complete and well summarized. The text is written correctly and readable with pleasure. The figures are nice and understandable.The only note that can be made is that the authors do not propose any hypothesis that explains the protective filter action that the placenta seems to do towards the fetus, as proved by the facts and by the cited literature.The placental decidua is known to contain many cells belonging to the innate immunity system. The abundance of natural killer (NK) cells (70%) and the interrelation between cytotrophoblasts and NK could be an explanation for the placental protective action against the viral invasion. The role of innate immune system in protecting fetuses and neonates from the infection by SARS-CoV- 2 has been reported (Carsetti R, Lancet 2020). Adding some elements on the physiology of the placenta from an immunological point of view and formulating hypotheses on the action of the innate immunity system would make the manuscript more appealing, while as appears, it is a very well done summary of the existing case histories. I would recommend to read the Pereira's manuscript (Annu Rev Virol. 2018) and to supplement the review with some element that increases interest in reading it.Concerning the table in supplemental files on my opinion it should be simplified.

Author Response

Our Responses to Reviewer 1’s Comments

As below, on behalf of my co-authors, I would like to clarify some of the points raised by the Reviewer. And we hope the Reviewer and the Editors will be satisfied with our responses to the ‘comments’ and the revision on the revised manuscript. Some detailed revisions were made with tracked changes in the manuscript.

The authors performed a very extensive review of the existing literature on the placental effects of SARS-CoV 2 infection acquired in pregnancy. The proposed documentation is very complete and well summarized. The text is written correctly and readable with pleasure. The figures are nice and understandable. The only note that can be made is that the authors do not propose any hypothesis that explains the protective filter action that the placenta seems to do towards the fetus, as proved by the facts and by the cited literature. The placental decidua is known to contain many cells belonging to the innate immunity system. The abundance of natural killer (NK) cells (70%) and the interrelation between cytotrophoblasts and NK could be an explanation for the placental protective action against the viral invasion. The role of innate immune system in protecting fetuses and neonates from the infection by SARS-CoV-2 has been reported (Carsetti R, Lancet 2020). Adding some elements on the physiology of the placenta from an immunological point of view and formulating hypotheses on the action of the innate immunity system would make the manuscript more appealing, while as appears, it is a very well done summary of the existing case histories. I would recommend to read the Pereira's manuscript (Annu Rev Virol. 2018) and to supplement the review with some element that increases interest in reading it.

Our Response: Thank you for the valuable comments. We agree that the maternal-fetal interface of placenta may act as a strong protective filter/barrier against infection, including SARS-CoV-2. A brief discussion on this was added, including the two references that were suggested. See page 2, lines 64 – 75 and references 10 and 11, page 11.

Concerning the table in supplemental files on my opinion it should be simplified.

Our Response: Thank you for the suggestion. We have simplified the supplementary Table by omitting some of the information on the pregnant mothers’ medical history. The in-text citations were also updated accordingly. See supplementary Table S1.

Reviewer 2 Report

Thank you for preparing this manuscript. The manuscript presents a comprehensive collation of available literature on the effect of Covid-19 on the placenta and the potential impact on pregnancy outcomes. I enjoyed reading the manuscript. The scientific content is sound but the manuscript can benefit from some revision and language editing. For instance:

line 56: which raises an important question on how successful is transplacental viral infection (intrauterine transmission) to the fetus.

line 60: what is preventing...

Line 150-132: although sound, are outside the focus of the review: please consider deleting.

Line 262: Should not be attributed: do you mean may not be attributed?

Line 289: Proceeded: do you mean preceded?

Line 292: remove were.

Line 322: exploited??

Line 325: were: was

Line 327: amniotic fluid ? samples? Line 328 remove 'in'.

Line 338: syncytiotrophoblast not --blastic cells.

Line 347: and the detrimental effects it Will bring? do you mean: may bring?

Line 353: There was .. thus far: replace with there are no...

Line 364: concluded?? meaning what?

Line 376: should read has not yet been...

Line 390: Collective informations on>>> please revise.

Author Response

Responses to Reviewer 2’s Comments

As below, on behalf of my co-authors, I would like to clarify some of the points raised by the Reviewer. And we hope the Reviewer and the Editors will be satisfied with our responses to the ‘comments’ and the revision on the revised manuscript. Some detailed revisions were made with tracked changes in the manuscript.

Thank you for preparing this manuscript. The manuscript presents a comprehensive collation of available literature on the effect of Covid-19 on the placenta and the potential impact on pregnancy outcomes. I enjoyed reading the manuscript. The scientific content is sound but the manuscript can benefit from some revision and language editing. For instance:

line 56: which raises an important question on how successful is transplacental viral infection (intrauterine transmission) to the fetus.

Our response:  Thank you for the comment. We have rephrased the sentence to “which raises an important question on the success rate of transplacental viral infection…” See page 2, line 55.

Line 60: what is preventing...

Our response:  Thank you for the comment. We have changed the sentence to “if this is true, the mechanism involved in preventing this highly infectious virus from reaching the fetus is still unclear”. See page 2, line 59.

Line 130-132: although sound, are outside the focus of the review: please consider deleting.

Our response:  As suggested, we have removed the sentence “The fatality rate of SARS-CoV-2 infection in pregnancy is no different from that of the general population,  and was relatively lower than SARS-CoV (15%) and MERS-CoV (27%)”. See page 4, line 142.

Line 262: Should not be attributed: do you mean may not be attributed?

Our response:  Thank you for pointing this out. We have changed the word “should” to “may”. See page 7, line 273.

Line 289: Proceeded: do you mean preceded?

Our response:  Thank you for the comment. We have rephrased the sentence to make it clearer “Placentas from two of the studies were not further tested for SARS-CoV-2”. See page 8, line 304.

Line 292: remove were.

Our response:  The word “were” was removed as suggested. See page 8, line 307.

Line 322: exploited??

Our response:  Thank you for identifying this mistake.  We have changed the word “exploited” to “reported”. See page 8, line 338.

Line 325: were: was

Our response:  Thank you for pointing this out. We have changed the word “were” to “was”. See page 8, line 341.

Line 327: amniotic fluid ? samples? Line 328 remove 'in'.

Our response:  The word “samples” was added after “amniotic fluid”. See page 8, line 340 and 343. The word “in” has been removed. See page 8, line 344.

Line 338: syncytiotrophoblast not --blastic cells.

Our response:  Thank you for pointing it out. We have changed the word to “syncytiotrophoblast”. See page 9, line 354.

Line 347: and the detrimental effects it Will bring? do you mean: may bring?

Our response:  We agree and have changed the word “will” to “may” as suggested. See page 9, line 363.

Line 353: There was .. thus far: replace with there are no...

Our response:  We have replaced the word “was” to “are” as suggested. See page 9, line 369.

Line 364: concluded?? meaning what?

Our response:  We have rephrased the sentence to “The histological features of SARS-CoV-2-infected placenta are not well defined hitherto…” See page 10, line 380.

Line 376: should read has not yet been...

Our response:  We have added the word “yet” as recommended. See page 10, line 392.

Line 390: Collective informations on>>> please revise.

Our response:  Thank you for the comment. We have rephrased the sentence “the cumulative published data on placenta with COVID-19 infection showed common histological features, which include”. See page 11, line 406.